# *Menura*: a code for simulating the interaction between a turbulent solar wind and solar system bodies

Etienne Behar[1,2], Shahab Fatemi[3], Pierre Henri[2,4], and Mats Holmström[2]

[1]Swedish Institute of Space Physics, Kiruna, Sweden
[2]Laboratoire Lagrange, Observatoire de la Côte d'Azur, Université Côte d'Azur, CNRS, Nice, France
[3]Department of Physics at Umeå University, Umeå, Sweden
[4]LPC2E, Orléans, France

**Correspondence:** E. Behar (etienne.behar@irf.se)

**Abstract.** Despite the close relationship between planetary science and plasma physics, few advanced numerical tools allow to bridge the two topics. The code *Menura* proposes a breakthrough towards the self-consistent modelling of these overlapping fields, in a novel 2-step approach allowing for the global simulation of the interaction between a fully turbulent solar wind and various bodies of the solar system. This article introduces the new code and its 2-step global algorithm, illustrated by a first example: the interaction between a turbulent solar wind and a comet.

## 1 Introduction

TEXT

For about a century, three main research fields have taken an interest in the various space plasma environments found around the Sun. On the one hand, two of them, namely planetary science and solar physics, have been exploring the solar system, to understand the functioning and history of its central star, and of its myriad of orbiting bodies. On the other hand, the third one, namely fundamental plasma physics, has been using the solar wind as a handy wind tunnel which allows researchers to study fundamental plasma phenomena not easily reproducible on the ground, in laboratories. During the last decades, the growing knowledge of these communities lead them to research on ever more overlapping topics. For instance, planetary scientists were initially studying the interaction between solar system bodies and a steady, ideally laminar solar wind, but they soon had to consider its eventful and turbulent nature to go further in the in situ space data analysis, further in their understanding of the interactions at various obstacles. Similarly, plasma physicists were originally interested in a pristine solar wind unaffected by the presence of obstacles. They however realised that the environment close to these obstacles could provide combinations of plasma parameters otherwise not accessible to their measurements in the unaffected solar wind. For a while now, we have seen planetary studies focusing on the effects of solar wind transient effects (such as Coronal mass Ejection CME or Co-rotational Interaction Region CIR) on planetary plasma environments, at Mars (Ramstad et al., 2017), Mercury (Exner et al., 2018), Venus (Luhmann et al., 2008) and comet 67P/C-G (Edberg et al., 2016; Hajra et al., 2018) to only cite a few, the effect of large scale fluctuations in the upstream flow on Earth's magnetosphere (Tsurutani and Gonzalez, 1987), and more generally the effect of solar wind turbulence on Earth magnetosphere and ionosphere (D'Amicis et al., 2020; Guio and Pécseli, 2021). Similarly,

plasma physicists have developed comprehensive knowledge of plasma waves and plasma turbulence in the Earth magnetosheath, presenting relatively high particle densities and electromagnetic field strengths, favourable for space instrumentation, and in a region more easily accessible to space probes than regions of unaffected solar wind (Borovsky and Funsten, 2003; Rakhmanova et al., 2021). More recently, the same community took an interest in various planetary magnetospheres, depicting plasma turbulence in various locations and of various parameters (Saur, 2021), and all references therein.

Various numerical codes have been used for the global simulation of the interaction between a laminar solar wind and solar system bodies, using MHD (Gombosi et al., 2004), hybrid (Bagdonat and Motschmann, 2002), or fully kinetic (Markidis et al., 2010) solvers. Similarly, solar wind turbulence in the absence of an obstacle has also been simulated using similar MHD (Boldyrev et al., 2011), hybrid (Franci et al., 2015), and fully kinetic (Valentini et al., 2007) solvers. In this context, we identify the lack of a numerical approach for the study of the interaction between a turbulent plasma flow (such as the solar wind) and an obstacle (such as a magnetosphere, either intrinsic or induced). Such a tool would provide the first global picture of these complex interactions. By shedding new lights on the long-lasting dilemma between intrinsic phenomena and phenomena originating from the upstream flow, it would allow invaluable comparisons between self-consistent, global, numerical results, and the worth of observational results provided by the various past, current and future exploratory space missions in our solar system.

The main points of interest and main questions motivating such a model can be organised as such:

- Macroscopic effects of turbulence on the obstacle

    - shape and position of the plasma boundaries (e.g. bow shock, magnetopause),

    - large scale magnetic reconnection,

    - atmospheric escape,

    - dynamical evolution of the magnetosphere.

- Microscopic physics and instabilities within the interaction region, induced by upstream turbulence

    - energy transport by plasma waves,

    - energy conversion by wave-particle interactions,

    - energy transfers by instabilities.

- The way incoming turbulence is processed by planetary plasma boundaries

    - sudden change of spatial and temporal scales,

    - change of spectral properties,

    - existence of a memory of turbulence downstream magnetospheric boundaries.

Indirectly, because of the high numerical resolution required to properly simulate plasma turbulence, this numerical experiment will provide an exploration of the various obstacles with the same high resolution in both turbulent and laminar runs, resolutions that have rarely been reached for planetary simulations, except for Earth's magnetosphere.

*Menura*, the new code presented in this publication, splits the numerical modelling of the interaction into two steps. Step 1 is a decaying turbulence simulation, in which electromagnetic energies initially injected at the large spatial scales of the simulation box cascades towards smaller scales. Step 2 uses the output of Step 1 to introduce an obstacle moving through this turbulent solar wind.

The code is written in `c++` and uses `CUDA` APIs for running its solver exclusively on multiple Graphics Processing Units (GPUs) in parallel. Section 2 introduces the solver, which is tested against classical plasma phenomena in Section 3. Sections 4 and 5 tackle the first and second step of the new numerical modelling approach, illustrating the decaying turbulence phase, and introducing the algorithm for combining the output of Step 1 together with the modelling of an obstacle (Step 2). Section 6 presents the first global result of *Menura*, providing a glimpse of the potential of this numerical approach, and introducing the forthcoming studies.

*Menura* source code is open source, available under the GNU General Public License.

## 2  The solver

In order to (i) achieve global simulations of the interactions while (ii) modelling the plasma kinetic behaviour, with regard to the computation capabilities currently available, a hybrid Particle-In-Cell (PIC) solver has been chosen for *Menura*. This well-established type of model resolves the Vlasov equation for the ions by discretising the ion distribution function as macroparticles characterized by discrete positions in phase space, and electrons as a fluid, with characteristics evaluated at the nodes of a grid, together with ion moments and electromagnetic fields. The fundamental computational steps of a hybrid PIC solver are:

- Particles' position advancement, or "push".

- Particles' moments mapping, or "gathering": density, current, eventually higher order, as required by the chosen Ohm's law.

- Electromagnetic field advancement, using either an ideal, resistive or generalised Ohm's law and Faraday's law.

- Particles' velocity advancement, or "push".

These steps are summarised in Figure 1. Details about these classical principles can be found in (Tskhakaya, 2008) and references therein. The bottleneck of PIC solvers is the particles' treatment, especially the velocity advancement and the moments computation (namely density and current). The simulation of plasma turbulence especially requires large amounts

of macro-particles per grid nodes. We therefore want to minimise both the amount of operations done on the particles and the number of particles itself. A popular method which minimises the amount of these computational passes through all particles is the Current Advance Method (CAM) (Matthews, 1994), for instance used for the hybrid modelling of turbulence by (Franci et al., 2015). Figure 1 presents *Menura*'s solver algorithm, built around the CAM, similar to the implementation of (Bagdonat and Motschmann, 2002). In this scheme, only four passes through all particles are performed, one position and one velocity pushes and two particle moments mappings. The second moment mapping in Figure 1, i.e. step 2, also produces the two pseudo-moments $\Lambda$ and $\Gamma$ used to advance the current as:

$$\Lambda = \sum_p \frac{q^2}{m} W(\mathbf{r}_{n+1}), \tag{1}$$

$$\Gamma = \sum_p \frac{q^2}{m} \mathbf{v}_{n+1/2} W(\mathbf{r}_{n+1}), \tag{2}$$

$$J_{n+1} = J_{n+1/2} + \frac{\Delta t}{2}(\Lambda \mathbf{E}^* + \Gamma \times \mathbf{B}), \tag{3}$$

with $\mathbf{E}^*$ the estimated electric field after the magnetic field advancement of step 4. $W(\mathbf{r}_{n+1})$ is the shape function, which attributes different weights for each node surrounding the macro-particle (Tskhakaya, 2008).

Central finite differences using a five-point stencil for evaluating derivatives as well as second order interpolations are used throughout the solver. The algorithm evaluates all fields values at the nodes (or equivalently cell-centres in this precise case) of the grid. In Appendix B, we discuss how such a scheme actually conserves $\nabla \cdot \mathbf{B} = 0$, as initially shown by (Tóth, 2000). Additionally, Appendix B illustrates the evolution of the total energy of the system.

The grid covering the physical simulation domain has an additional 2-node wide band, the guard or ghost nodes, allowing to solve derivatives using (central) finite differences at the very edge of the physical domain. For periodic boundary conditions, as used along all directions during Step 1 of the simulation, the value at the opposite edge of the physical domain are copied to the guard nodes. Other boundary conditions will be discussed later when introduced.

The mapping of the particle moments are done using an order-two, triangular shape function: one macro-particle contributes to 9 grid nodes in 2D space ( respectively 27 in 3D space), using 9 (respectively 27) different weights. The interpolation of the field values from the nodes to the macro-particles' positions uses the exact same weights, with 9 (respectively 27) neighbouring nodes contributing to the fields values at a particle position.

As illustrated in Figure 1, the position and velocity advancements are done at interleaved times, similarly as a classical second order leap-frog scheme. However, since the positions of the particles are needed to evaluate their acceleration, the CAM scheme is not strictly speaking a leap-frog integration scheme. Another difference in this implementation is that velocities are advanced using the Boris method (Boris, 1970).

The Ohm's law is at the heart of the hybrid modelling of plasmas. *Menura* uses the following form of the law, here given in SI units. In this formulation, the electron inertia is neglected, and the quasi-neutral approximation $n \sim n_i \sim n_e$ is used

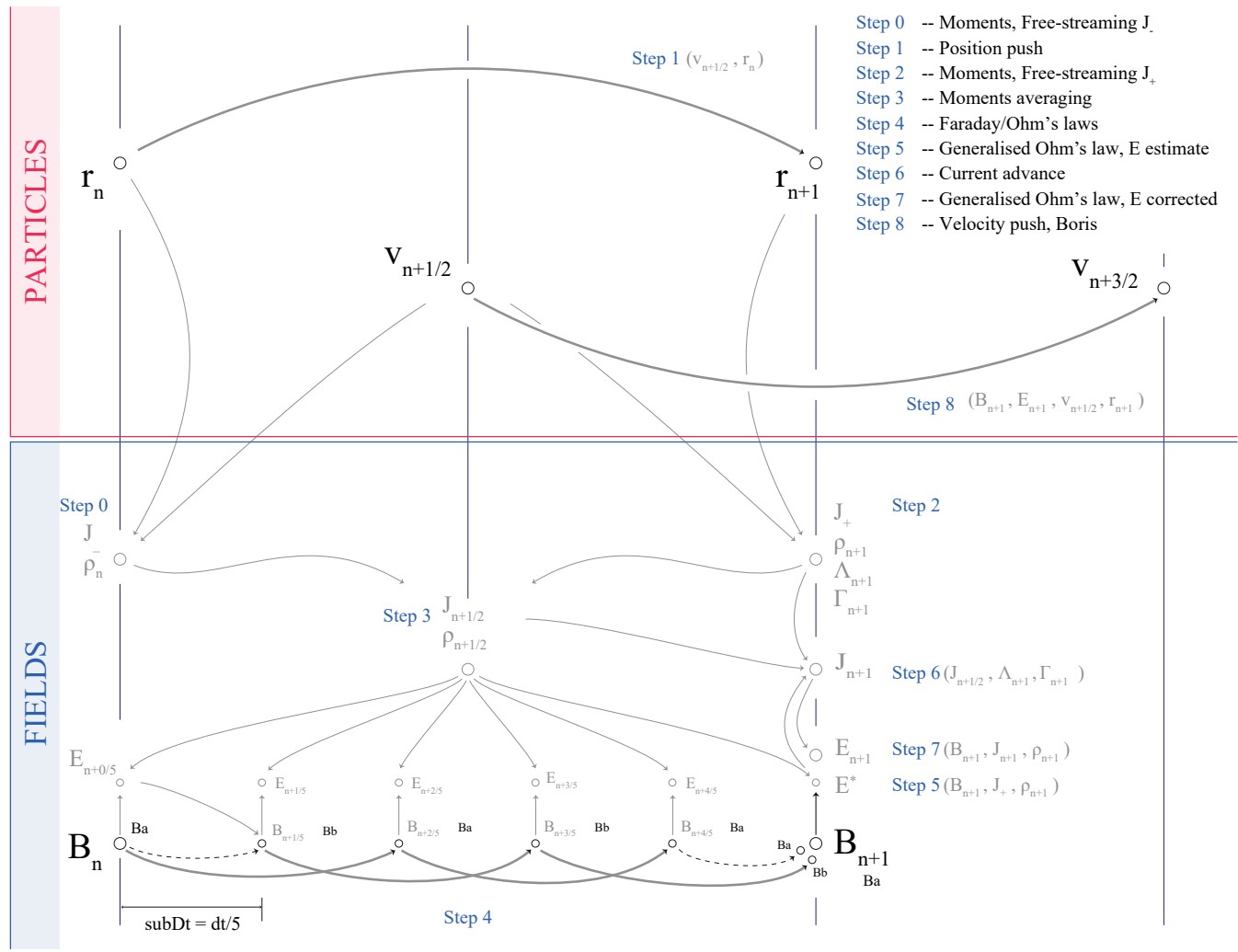

**Figure 1.** Algorithm of *Menura*'s solver, with its main operations numbered from 0 to 8, as organised in the `main` file of the code. **r** and **v** are the position and velocity vectors of the macro particles. Together with the magnetic field **B**, they are the only variables necessary for the time advancement. The electric field **E**, the current **J**, the charge density $\rho$, as well as the CAM pseudo-moments $\Lambda$ and $\Gamma$, are obtained from **r**, **v** and **B**.

(Valentini et al., 2007). Additionally, neglecting the time derivative of the electric field in the Ampere-Maxwell's law (Darwin's hypothesis), one gets the total current through the curl of the magnetic field. This formulation highlights the need for only three types of variables to be followed through time, namely the magnetic field, and the particles position and velocity, while all other variables can be reconstructed from these three.

$$120 \quad \mathbf{E} = -\mathbf{u_i} \times \mathbf{B} + \frac{1}{en}\mathbf{J} \times \mathbf{B} - \frac{1}{en}\nabla \cdot p_e - \eta_h \nabla^2 \mathbf{J} \tag{4}$$

The Faraday's law is used for advancing the magnetic field in time:

$$\frac{\partial \mathbf{B}}{\partial t} = -\nabla \times \mathbf{E} \tag{5}$$

The electron pressure is obtained assuming it results from a polytropic process, with an arbitrary index $\kappa$, to be chosen by the user. In all the results presented below, an index of 1 was used, corresponding to an isothermal process.

$$125 \quad p_e = p_{e0}\left(\frac{n_e}{n_{e0}}\right)^{\kappa} \tag{6}$$

Using much less memory than the particles' variables, the fields can be advanced in time using a smaller time step and another leap-frog-like approach, as illustrated in Figure 1, step 4 (Matthews, 1994).

Additional spurious high-frequency oscillations are the default behaviour of finite differences schemes. Two main families
of methods are used to filter out these features, the first being an additional step of field smoothing, the second using the direct inclusion of a diffusive term in the differential equation of the system, acting as a filter (Maron et al., 2008). For *Menura*, we have retained the second approach, implementing a term of hyper-resistivity in the Ohm's law, introducing the Laplacian of the total current and the hyper-resistivity coefficient, $\eta_h \nabla^2 \mathbf{J}$. The dissipative scale $L_{\mathrm{dis}}$ of such a term is characterised by the physical time of the simulation $T = $ nb. iterations $\times\, dt$ and the resistivity, such as $L_{\mathrm{dis}} = (\eta_h \cdot T)^{1/4}$.

The stability of hybrid solvers is sensitive to low ion densities. We use a threshold value equal to a few percent of the background density, 5% in the following examples, threshold below which a node is considered as a vacuum node, and only the resistive terms of the generalised Ohm's law of Equation 4 are solved using a higher value of resistivity $\eta_{h\ \mathrm{vacuum}}$ (Holmström, 2013). This way, terms proportional to $1/n$ do not exhibit nonphysical values where the density may get locally very low, due
to the thermal noise of the PIC macro-particle discretisation.

All variables in the code are normalised using the background magnetic field amplitude $B_0$ and the background plasma density $n_0$. All variables are then expressed in terms of either these two background values, or equivalently in terms of the

| | |
|---|---|
| $B_0$ | $B_0$ |
| $n_0$ | $n_0$ |
| $v_0$ | $v_{A0} = B_0/\sqrt{\mu_0 m_i n_0}$ |
| $\omega_0$ | $\omega_{ci0} = eB_0/m_i$ |
| $x_0$ | $d_{i0} = v_{A0}/\omega_{ci0}$ |
| $t_0$ | $1/\omega_{ci0}$ |
| $E_0$ | $v_{A0} \cdot B_0$ |
| $p_0$ | $B_0^2/(2\mu_0)$ |
| $m_0$ | $m_i n_0 x_0^3$ |
| $q_0$ | $e\, n_0 x_0^3$ |

**Table 1.** Background values used to normalise all variables in the solver (cf. Eq. 7).

proton gyrofrequency $\omega_{ci0}$ and the Alfven velocity $v_{A0}$. We define normalised variables $\tilde{a}$ as obtained by dividing its physical value by its "background" value:

$$\tilde{a} = \frac{a}{a_0} \tag{7}$$

All background values are given in Table 1, and the normalised equations of the solver are given in Appendix A.

## 3 Physical tests

In this section, the code is tested against well-known, collisionless plasma processes, and their solutions given by the linear full kinetic solver *WHAMP* (Rönnmark, 1982). A polytropic index of 1 is used here, with no resistivity. We first explore MHD scales, simulating Alfvénic and magnetosonic modes. We use a 2-dimensional spatial domain with one preferential dimension chosen as x. A sum of six cosine modes in the component of the magnetic field along the $x$-direction direction are initialised, corresponding to the first six harmonics of this periodic box. The amplitude of these modes is 0.05 times the background magnetic field $B_0$, which is taken either along (Alfvén mode) or across (magnetosonic mode) the propagation direction x. Data are recorded along time and along the main spatial dimension x (saving one cut, given by one single index along the $y$-direction), resulting in the 2D field $B(x,t)$. The 2-dimensional Fourier transform of this field is given in Figure 2 (Alfvénic fluctuations to the left, magnetosonic to the right). On this $(\omega, k)$-plane, each mode can be identified as a point of higher power, six points for six initial modes. The solutions given by *WHAMP* for the same plasma parameters are shown by the solid lines, and a perfect match is found between the two models. Close to the ion scale $k \cdot d_{i0} = 1$, *WHAMP* and *Menura* display two different branches that originate from the Alfvén mode, splitting for higher frequencies into the whistler and the ion cyclotron branches. The magnetosonic modes were also tested using a different polytropic index of 5/3 instead of 1, resulting in a shift of the dispersion relation along the $\omega$-axis. Changing the polytropic index in both *Menura* and *WHAMP* resulted in the same agreement.

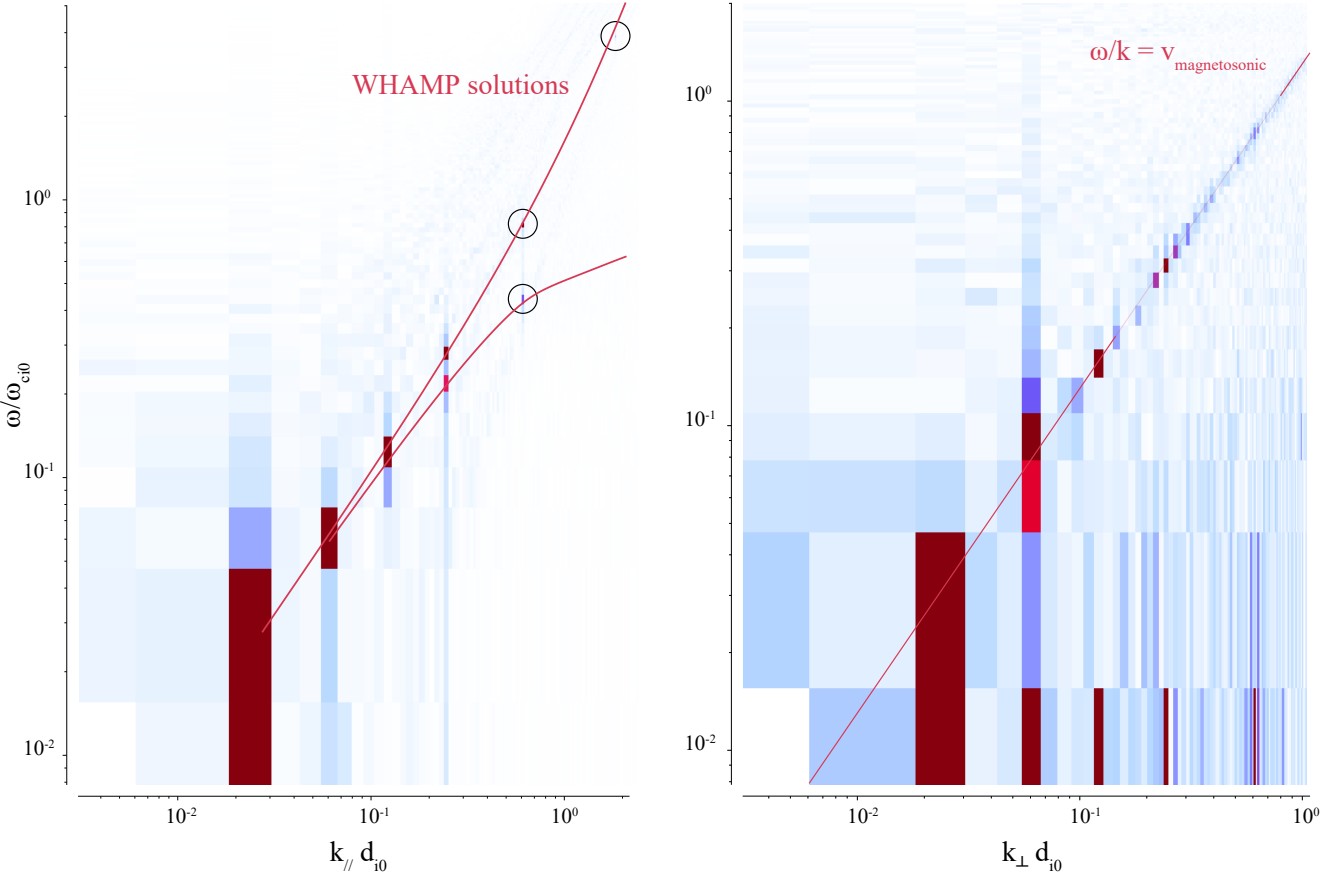

**Figure 2.** MHD modes dispersion relations, as solved by *WHAMP* and *Menura*. $B_0 = 1.8$ nT, $n_0 = 1.$ cm$^{-3}$, $T_{i0} = 10^4$ K, $T_{e0} = 10^5$ K. Left-hand panel: Alfvénic modes, $B_0$ taken along the main spatial dimension. Right-hand panel: Magnetosonic modes, $B_0$ taken perpendicular to the main spatial dimension.

With the MHD scales down to ion inertial scales now validated, we explore the ability of the solver to account for further ion kinetic phenomena, first with the classical case of the two-stream instability (also known as the ion-beam instability, given the following configuration). Two Maxwellian ion beams are initialised propagating with opposite velocities along the main dimension **x**. A velocity separation of $15v_{th}$ is used in order to excite only one unstable mode. The linear kinetic solver *WHAMP* is used to identify the expected growth rate associated to the linear phase of the instability, before both beams get strongly distorted and mixed in phase space during the nonlinear phase of the instability (not captured by WHAMP). During this linear phase, *Menura* results in a growing circularly polarised wave, and the amplitude's growth of the wave is shown in Figure 3. Both growth rates match perfectly.

Finally, we push the capacities of the model to the case of the damping of an ion acoustic wave through Landau resonance. A very high number of macro-particles per grid node is required to resolve this phenomenon, so enough resonant particles take

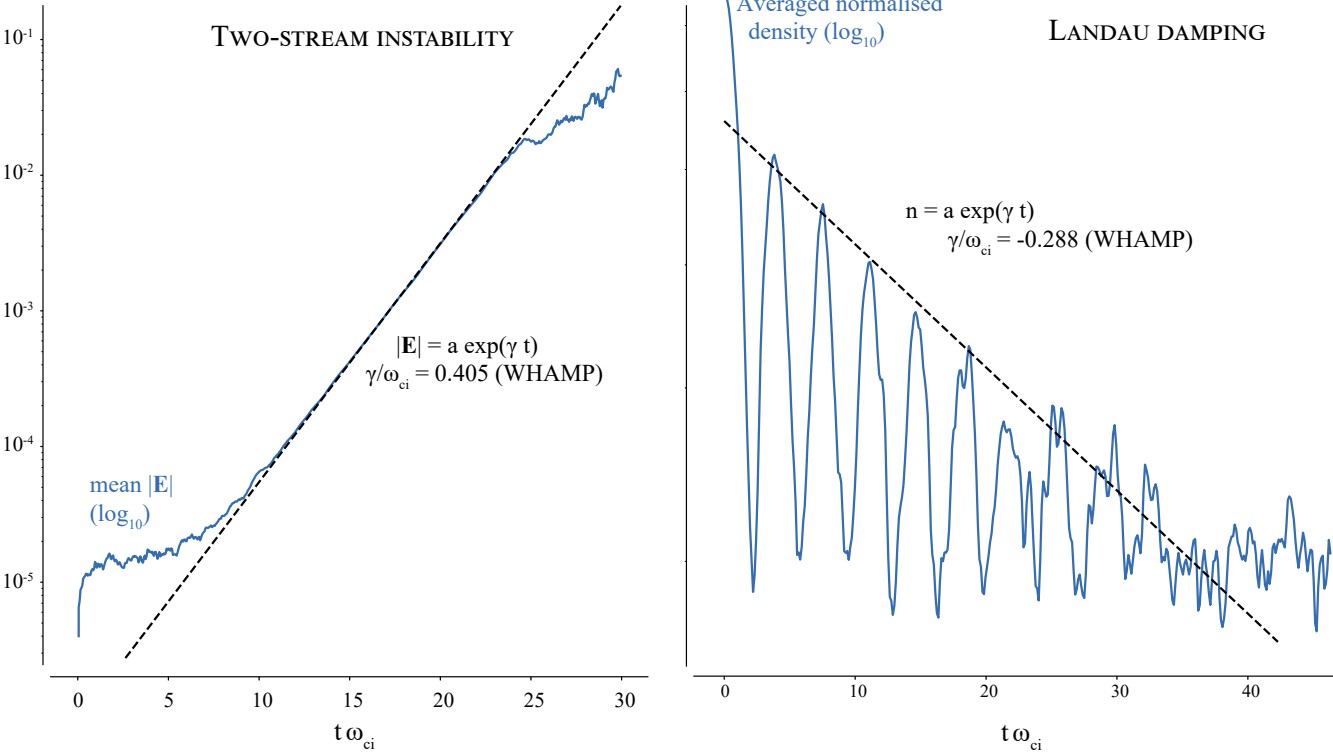

**Figure 3.** Left-hand side, growth during the linear phase of the ion-ion two-stream instability; Right-hand side, Landau damping of an ion acoustic mode. Two-stream instability: $B_0 = 1.8$ nT, $n_0 = 1.$ cm$^{-3}$, $T_{i0} = 10^2$ K, $T_{e0} = 10^3$ K. Landau Damping: $B_0 = 1.8$ nT, $n_0 = 5.$ cm$^{-3}$, $T_{i0} = 1.5 \cdot 10^4$ K, $T_{e0} = 10^5$ K

part in the interaction with the wave. The amplitude of the initial, single acoustic mode is taken as 0.01 times the background density, taken along the main spatial dimension of the box. This low amplitude, allowing for comparison with the linear solver,
further increase the need for a high number of particle per node, so the 1% oscillation in number density can be resolved by the finite number of particles. For this run, 32768 ($2^{15}$) particles per grid node were used. The decrease in the density fluctuation through time, spatially averaged, is shown in Figure 3, with again the corresponding solution from *WHAMP*. A satisfying agreement is found during the first 6 oscillations, before the noise in the hybrid solver output (likely associated to the macroparticle thermal noise) takes over. Admittedly, the amount of particle per node necessary to well resolve this phenomenon
is not practical for the global simulations which *Menura* (together with all global PIC simulations) aims for.

For the classical tests presented above, spanning over MHD and ion kinetic scales tests, *Menura* agrees with theoretical and linear results. In the next section, the simulation of a decaying turbulent cascade provides one final physical validation of the solver, through all these scales at once.

## 4 Step 1: Decaying turbulence

We use *Menura* to simulate plasma turbulence using a decaying turbulent cascade approach: at initial time $t = 0$, a sum of sine modes with various wave vectors $\mathbf{k}$, spanning over the largest spatial scales of the simulation domain, are added to both the homogeneous background magnetic field $\mathbf{B}_0$ and the ion bulk velocity $\mathbf{u}_i$. Particle velocities are initialised according to a Maxwellian distribution, with a thermal speed equal to one Alfvén speed, and a bulk velocity given by the initial fluctuation. Without any other forcing later-on, this initial energy cascades, as time advances, towards lower spatial and temporal scales,

forming vortices and reconnecting current sheets (Franci et al., 2015). Using such Alfvénic perturbation is motivated by the predominantly Alfvénic nature of the solar wind turbulence measured at 1 au (Bruno and Carbone, 2013).

In this 2-dimensional set-up, $\mathbf{B}_0$ is taken along the $\mathbf{z}$-direction, perpendicular to the simulated spatial domain $(\mathbf{x}, \mathbf{y})$, whereas all initial perturbations are defined within the simulation plane. Their amplitude is 0.5 $B_0$, while their wave vectors are taken with values between $k_{\text{inj, min}} = 0.01 \, d_{i0}^{-1}$ and $k_{\text{inj, max}} = 0.1 \, d_{i0}^{-1}$, so energy is only injected in MHD scales, in the inertial range

(Kiyani et al., 2015). Because we need these perturbation fields to be periodic along both directions, the $k_x$ and $k_y$ of each mode corresponds to harmonics of the simulation box dimensions. Therefore, a finite number of wave vector directions is initialised. Along these constrained directions, each mode in both fields has two different, random phases. The magnetic field is initialised such that is it divergence-free.

For this example, the box is chosen to be 500 $d_{i0}$ wide on both dimensions, subdivided by a grid $1000^2$ nodes wide. The

200 corresponding $\Delta x$ is 0.5 $d_{i0}$, and spatial frequencies are resolved over the range $[0.0062, 6.2] \, d_{i0}^{-1}$. The time step is 0.05 $\omega_{ci0}^{-1}$. 2000 particles per grid node are initialised with a thermal speed of 1 $v_A$. The temperature is isotropic and a plasma beta of 1 is chosen for both the ion macro-particles and the electronic massless fluid. The polytropic index is 1 and a normalised hyper-resistivity of $\eta_h = 2 \cdot 10^{-3}$ is used, corresponding to a dissipative scale at time $t = 500 \, \omega_{ci0}^{-1}$ of 1. $d_{i0}$, i.e. the scale of the smallest fluctuations simulated with a node spacing of $\Delta x = 0.5 d_{i0}$.

At time $t = 500 \, \omega_{ci0}^{-1}$, the perpendicular (in-plane) fluctuations of the magnetic field have reached the state displayed in Figure 4, left-hand panel. Vortices and current sheets give a maximum $B_\perp / B_0$ of about 1, a result consistent with solar wind turbulence observed at 1 au (Bruno and Carbone, 2013). The omni-directional power spectra of both the in-plane magnetic field fluctuations and the in-plane ion bulk velocity fluctuations are shown in the right-hand panel of the same figure. Omni-directional spectra are computed as follows, with $\hat{f}$ the (2D) Fourier transform of $f$:

$$P_f(k_x, k_y) = |\hat{f}|^2 \tag{8}$$

These spectra are not further normalised and are given in arbitrary units. We then compute a binned statistics over this 2-dimensional array to sum up its values within the chosen bins of $k_\perp$, which correspond to rings in the $(k_x, k_y)$-plane. The

| | |
|---|---|
| $B_0$ | 2.5 nT |
| $n_0$ | 1 cm$^{-3}$ |
| $\omega_{ci0}$ | 0.24 s |
| $d_{i0}$ | 228 km |
| $v_{A0}$ | 55 km/s |
| $v_{thi0}$ | 55 km/s |
| $\beta_{i0} = \beta_{e0}$ | 1 |
| $B_{\perp 0}/B_0$ | 0.7 |

**Table 2.** Initial parameters of the decaying turbulence run

width of the rings, constant through all scales, is arbitrarily chosen so the resulting 1-dimensional spectrum is well resolved (not too few bins), and not too noisy (not too many bins).

$$P_f(k_\perp) = \sum_{k_\perp \in [k_{\perp 0},\ k_{\perp 0} + \delta k_\perp]} |\hat{f}|^2 \tag{9}$$

For a vector field such as $\mathbf{B}_\perp = (B_x, B_y)$, the spectrum is computed as the sum of the spectra of each field component:

$$P_{\mathbf{B}_\perp}(k_\perp) = P_{B_x}(k_\perp) + P_{B_y}(k_\perp). \tag{10}$$

The perpendicular magnetic and kinetic energy spectra exhibit power laws over the inertial (MHD) range consistent with spectral indexes -5/3 and -3/2, respectively, between $k_{\text{inj max}} = 0.1\ d_{i0}^{-1}$ and break points around 0.5 $d_{i0}^{-1}$. We remind that a spectral index -5/3 is consistent with the Goldreich-Sridhar strong turbulence phenomenology (Goldreich and Sridhar, 1997) that leads to a Kolmogorov-like scaling in the plane perpendicular to the background magnetic field, while a spectral index -3/2 is consistent with the Iroshnikov-Kraichnan scaling (Kraichnan, 1965). These spectral slopes are themselves consistent with observations of magnetic and kinetic energy spectra associated with solar wind turbulence (Podesta et al., 2007; Chapman and Hnat, 2007). For higher wavenumbers, both spectral slopes get much steeper, and after a transition region within [0.5, 1.] $d_{i0}^{-1}$ get to a value of about -3.2 and -4.5 when reaching the proton kinetic scales for respectively the perpendicular magnetic and kinetic energies, consistent with spectral index found at sub-ion scales by previous authors (Franci et al., 2015; Sahraoui et al., 2010, e.g.). Additionally, the initial spectra of the magnetic field and bulk velocity perturbations are over-plotted, to show where the energy is injected in the lower spatial frequencies (using the magnetic field fluctuations), and the level of noise introduced by the finite number of particles per node used, at high frequencies (using the bulk velocity field).

## 5 Step 2: Obstacle

*Menura* has shown satisfactory results on plasma turbulence, over three orders of magnitude in wavenumbers. We now start the second phase of the simulation, resuming it at $t = 500\ \omega_{ci0}^{-1}$, corresponding to the snapshot studied in the previous section. We

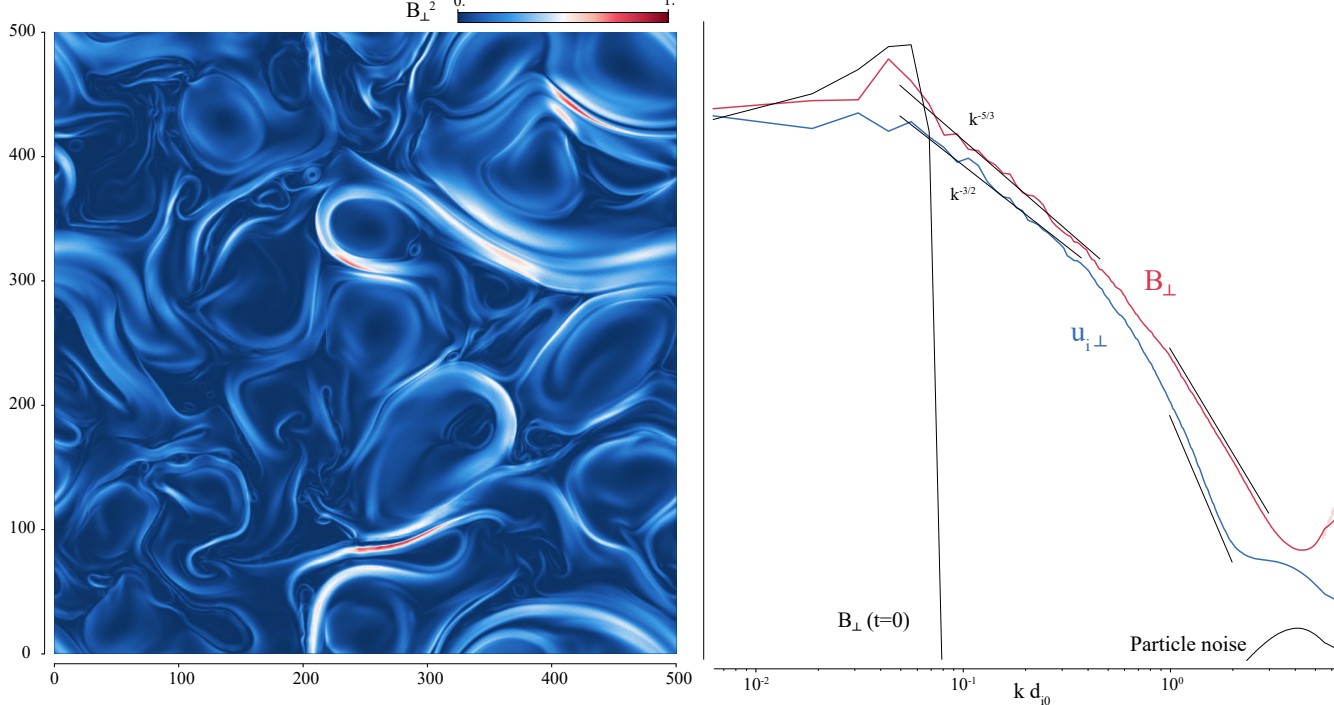

**Figure 4.** Decaying turbulence at time $500\ \omega_{ci0}^{-1}$. The left-hand panel shows the squared in-plane (perpendicular) magnetic field amplitude, while the right-hand panel presents the omnidirectional power density spectra of the same perpendicular magnetic field as well as the perpendicular velocity field.

keep *all* parameters unchanged (including the polytropic index of 1 and the hyper-resistivity of $1.5 10^{-3}$), but add an obstacle with a relative velocity with regard to the frame used in the first phase, evolving through this developed turbulence. Particles
and fields are advanced with the exact same time and spatial resolutions as previously, so the interaction between this obstacle and the already-present turbulence is solved with the same self-consistency as in the first phase, with only one ingredient added: the obstacle.

## 5.1 A comet

This obstacle is chosen here to be an intermediate activity comet, meaning that its neutral outgassing rate is typical of an icy
nucleus at a distance of about 2 astronomical units from the Sun. A comet nucleus is from a few to a few tens of kilometers large, with a gravitational pull not strong enough to overcome the kinetic energy gained by the molecules during sublimation. Comprehensive knowledge on this particular orbital phase of comets has recently been generated by the European *Rosetta* mission, which orbited its host comet for two years (Glassmeier et al., 2007). The first and foremost interest of such an object for this study is the size of its plasma environment, which can be evaluated using the gyroradius of water ions in the solar
wind at 2 au. The expected size of the interaction region is about 4 times this gyro-radius (Behar et al., 2018), and with

the characteristic physical parameters of Table 2, the estimated size of the interaction region is 480 $d_{i0}$. In other words, the interaction region spans exactly over the range of spatial scales probed during the first phase of the simulation, including MHD and ion kinetic scales.

The second interest of a comet is its relatively simple numerical implementation. Considering the spatial resolution of the simulation, the solid nucleus can be neglected. By also neglecting the gravitational force on molecules as well as any intrinsic magnetic field, the obstacle is only made of cometary neutral particles being photo-ionised *within* the solar wind. Over the scales of interest for this study, the neutral atmosphere can be modelled by a $1/r^2$ radial density profile, and considering the coma to be optically thin, ions are injected in the system with a rate following the same profile. This is the Haser Model (Haser, 1957), and simulating a comet over scales of hundreds of $d_{i0}$ only requires to inject cold cometary ions at each time step with the rate

$$q_i(r) = \nu_i \cdot n_0(r) = \frac{\nu_i Q}{4\pi u_0 r^2}, \tag{11}$$

with $r$ the distance from the comet nucleus of negligible size, $\nu_i$ the ionisation rate of cometary neutral molecules, $n_0$ the neutral cometary density, $Q$ the neutral outgassing rate, $u_0$ the radial expansion speed of the neutral atmosphere.

One additional simplification is to limit the physico-chemistry of the cometary environment to photo-ionisation, thus neglecting charge exchanges between the solar wind and the coma, as well as electron impact ionization. Both processes can significantly increase the ionisation of the neutral coma (Simon Wedlund, Cyril et al., 2019). A global increase or a local change in the production profile is not expected to impact the initial main goal of the model, which is to simulate the turbulent nature of the solar wind during its interaction with an obstacle. We note however that the influence of upstream turbulence on the physico-chemistry of an obstacle is yet another promising prospect for the code.

## 5.2 Reference frame

The first phase of the simulation, the decaying turbulence phase, was done in the plasma frame, in which the average ion bulk velocity is 0. Classically, planetary plasma simulations are done in the planet reference frame: the obstacle is static and the wind flows through the simulation domain. In this case, a global plasma reference frame is most of the time not defined. In *Menura*, we have implemented the second phase of the simulation – the interaction phase – in the exact same frame as the first phase, which then corresponds to the plasma frame of the upstream solar wind, before interaction. In other words, the turbulent solar wind plasma is kept "static", and the obstacle is moving through this plasma. The reason motivating this choice is to keep the turbulent solar wind "pristine", by continuing its resolution over the exact same grid as in phase one. Another motivation for working in the solar wind reference frame is illustrated in Figure 5, in which we compare the exact same simulation done in each frame, using a laminar upstream flow. If the macroscopic result remains unchanged between the two frames, we find strong small scale numerical artifacts propagating upstream of the interaction in the comet reference frame, absent in the solar wind reference frame. Small scale oscillations are common in hybrid PIC simulations, and are usually filtered with either resistivity and/or hyper-resistivity, or with an ad-hoc smoothing method. Note that none of these methods are used in the

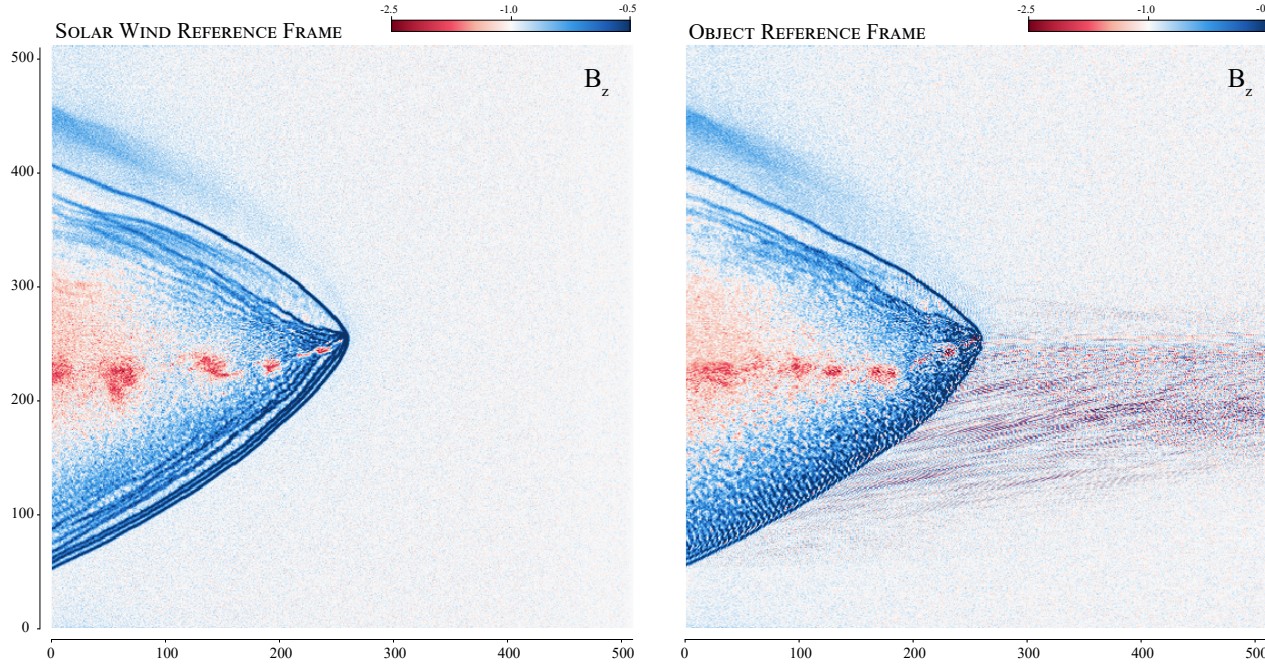

**Figure 5.** The interaction between a comet and a laminar flow, in the object rest frame (right) and the upstream solar wind reference frame (left). The magnetic field amplitude is shown.

present example. We demonstrate here the role of the reference frame in the production of one type of small scale oscillations, and ensure that their influence over the spectral content of upstream turbulence is minimised, already without the implemented hyper-resistivity.

To summarise, by keeping the same reference frame during Step 1 and 2, the only effective difference between the two phases is the addition of sunward moving cometary macroparticles.

Another major advantage of working in the solar wind reference frame is the possibility to simulate magnetic field variations in all directions, including the relative plasma-object direction. For studying the interaction between Co-rotating Interaction Regions and an object for instance, one need to vary the direction of the magnetic field upstream of the object of interest. In the object reference frame, such a temporal variation of the magnetic field is frozen-in the flow and advected downstream through the simulation domain by the convective electric field. Considering the ideal Ohm's law $\mathbf{E} = -\mathbf{u}_i \times \mathbf{B}$ and Faraday's law $\partial_t \mathbf{B} = -\nabla \times \mathbf{E}$, and considering a plasma flowing along the $x$-axis $\mathbf{u}_i = u_i \hat{x}$, we get the time evolution of each magnetic field component as

$$\partial_t B_x = \partial_y(u_i B_y) - \partial_z(u_i B_z)$$

$$\partial_t B_y = -\partial_x(u_i B_y)$$

$$\partial_t B_z = \partial_z(u_i B_z)$$

The direct implication of this system is that any temporal variation we may force on the upstream $B_x$ cannot have a self-consistent influence on the time evolution of the magnetic field elsewhere: only variations forced along magnetic field components perpendicular to the flow direction can be advected downstream, through this ideal frozen-in condition. In contrast, when working in the solar wind reference frame, we can impose spatial fluctuations of the magnetic field (equivalently temporal in the object frame) in all directions: in this frame these fluctuations are not being advected, it is rather the object itself mov-

ing through the fluctuations. This effectively removes the constraint on flow-aligned variation of the magnetic field, opening promising possibilities for the simulation of various solar wind events, such as CIRs or sector boundary crossings.

### 5.3    Algorithm

By working in the solar wind reference frame, the obstacle is moving within the simulation domain. Eventually, the obstacle would reach the boundaries of the box, before steady-state is reached. We therefore need to somehow keep the obstacle close

to the centre of the simulation domain. This is done by shifting all particles and fields of $n\Delta x$ every $m$ iterations, $n, m \in \mathbb{N}$, as illustrated in Figure 6. Using integers, the shift of the field is simply a side-way copy of themselves without the need of any interpolation, and the shift of the particles is simply the subtraction of $n\Delta x$ to their $x$-coordinate. Field values as well as particles ending up downstream of the simulation domain are discarded.

This leaves only the injection boundary to be dealt with. There, we simply inject a slice of fields and particles picked from the

output of Step 1, using the right slice index in order to inject the continuous turbulent solution, as shown in Figure 6. These slices are $n\Delta x$ wide.

     With `idx_it` the index of the iteration, the algorithm illustrated in Figure 6 is then:

- Inject cometary ions according to $q_i(r)$ (cf. Eq. 11)

- Advance particles and fields (cf. Figure 1)

- If `idx_it%m=0`

  - Shift particles and fields of $-\mathrm{n}\Delta\mathrm{x}$

  - Discard downstream values

  - Inject upstream slice `idx_slice` from Step 1 output

  - Increment `idx_slice`

– Increment `idx_it`

This approach has one constraint, we cannot fine-tune the relative speed $v_0$ between the wind and the obstacle, which has to be

$$v_0 = \frac{n}{m} \frac{\Delta x}{\Delta t} \tag{12}$$

in order for the obstacle to come back to its position every `m` iterations, and therefore not drift up- or downstream of the
simulation domain.

## 5.4 CUDA and MPI implementation, performances

The computation done by *Menura*'s solver (Figure 1) is entirely executed on multiple GPUs (Graphics Processing Units),
written in `c++` in conjunction with the `CUDA` programming model and the `MPI` standard, which allows to split the problem
and distribute it over multiple cards (i.e. processors). GPUs can run simultaneously thousands of threads, and can therefore
tremendously accelerate such applications. The first implementation of a hybrid-PIC model on such devices was done by
(Fatemi et al., 2017). However their still limited memory (up to 80GB at the time of writing) is a clear constraint for large
problems, especially for turbulence simulation which requires a large range of spatial scales *and* a very large number of particles
per grid node. The use of multiple cards becomes then unavoidable, and the communications between them is implemented
using a CUDA-aware version of MPI, the Message Passing Interface. The division of the simulation domain in the current
version of *Menura* is kept very simple, with equal size rectangular sub-domains distributed along the direction perpendicular
to the motion of the obstacle: one sub-domain spans the entire domain along the x-axis with its major dimension, as shown
in Figure 6. MPI communications are done for particles after each position advancement, and for fields after each solution
of the Ohm's law and the Faraday's law. But since the shift of fields and particles described in the previous section is done
purely along the obstacle motion direction, no MPI communication is needed after the shifts, thanks to the orientation of the
sub-domains.

Another limitation in using GPUs is the data transfer time between the CPU and the cards. In *Menura*, all variables are
initialised on the CPU, and are saved from the CPU. Data transfers are then unavoidable, before starting the main loop, and
every time we want to save the current state of the variables. During Step 2 of the simulation, a copy of the outputs of Step 1 is
needed, which effectively doubles the memory needed for Step 2. This copy is kept on the CPU (in the `tank` object) in order
to make the most out of the GPUs memory, in turn implying that more CPU-GPU communications are needed for this second
step. Every time we inject a slice of fields and particles upstream of the domain, only this amount of data is copied from the
CPU to the GPUs, using the `injector` data structure as sketched in Figure 6.

SIMULATING IN SOLAR WIND FRAME            SHIFTING FIEDLS AND PARTICLES

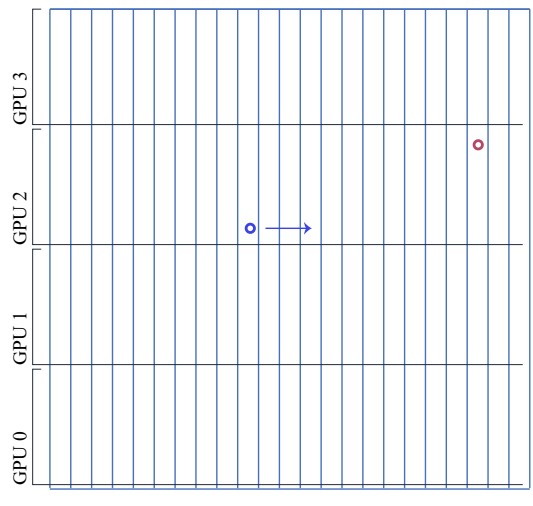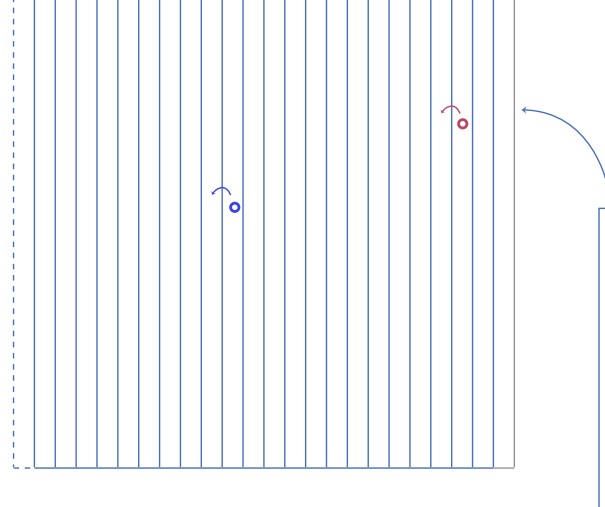

○ Planetary ion, v= Δx/(n·Δt)

○ Solar wind ion, v=0

INJECTOR

TANK
(output of Step 1)

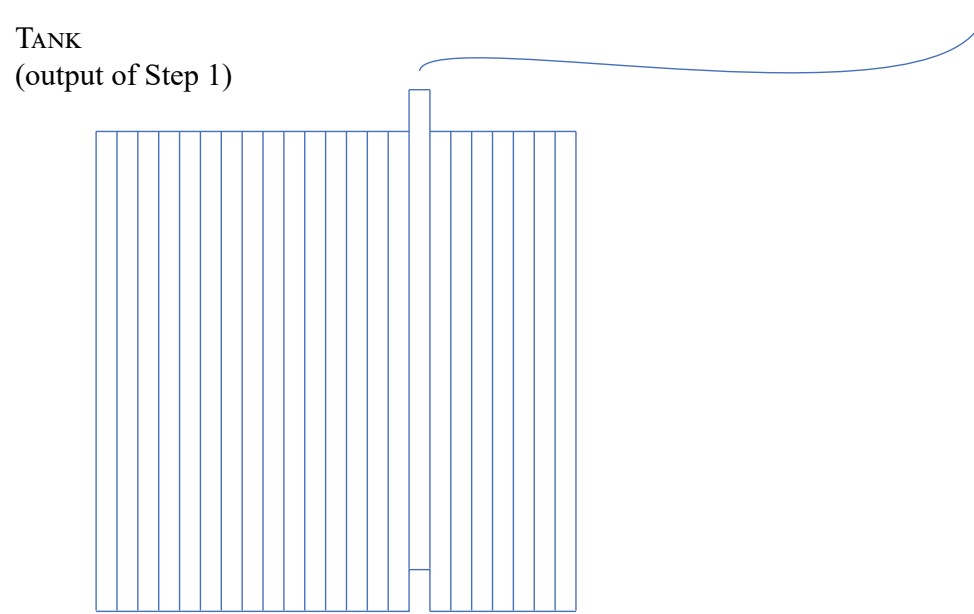

**Figure 6.** Injection algorithm for simulating a moving object within the simulation domain.

### 5.4.1 Profiling

350 For Step 1, the decaying turbulence ran 10000 iterations, four NVIDIA V100 GPUs were used with 16GB memory each, corresponding to one complete node of the IDRIS cluster Jean Zay. A total of 2 billion particles (500 million per card) were initialised. The time for the solver on each card reached a bit more than three hours, with a final total coast of about 13 hours of computation time for this simulation, taking into account all four cards, and the variables initialisation and output. Step 2 was executed on larger V100 32GB cards, providing much more room for the addition of 60000 cometary macro-particles per 355 iteration.

 During Step 1, 87.3% of the computation time was spent on moments mapping, i.e. steps 0 and 2 in the algorithm of Figure 1, while respectively 2.7% and 0.8% were spent on advancing the particles velocity and position. The computation of the Ohm's and Faraday's laws sums up to 0.5%. 0.9% was utilised for MPI communications of field variables, while only 0.08% was dedicated for particles MPI communications, due to the limited particle transport happening in Step 1.

360 91% of the total solver computation time is devoted to particles treatment, with 96% of that part spent on particles moment mapping, which might seem a suspiciously large fraction. We note however that such a simulation is characterised by its large number of particles per node, 2000 in our case. 99.6% of the total allocated memory is devoted to particles. The time spent to map the particles on the grid is also remarkably larger than the time spent to update their velocity, despite both operations being based on the same interpolation scheme. However, during the mapping of the particles moments, thousands of particles need 365 to *increment* the value at particular memory addresses (corresponding to macro-particle density and flux), whereas during the particle velocity advancement, thousands of particles only need to *read* the value of the same addresses (electric and magnetic field).

### 5.4.2 Scalability

An important part of performance testing for parallelised codes is the scalability of the parallelisation. When each parallel 370 process is serial (i.e. one thread for one process), the strong and weak scaling of the code are classical performance tests, with theoretical laws available for comparison (respectively the Amdahl's and Gustafson's law). These laws cannot be directly adapted to the case of devices that already have a highly parallel structure, such as GPUs. We however can approach the same strong and weak scaling properties of the code to get some valuable insights on the performance of the *Menura*'s MPI implementation.

375 Strong scaling refers to the speed-up (gain of computation time) obtained while simulating the same problem (same grid size and same amount of particle per grid node) with a growing number of processes (the load thus decreasing on each GPU). The upper panel of Figure 7 shows the speed-up obtained using from 1 to 25 processes (i.e. GPUs) solving the same problem: an homogeneous, fully periodic, 2D plasma box, with no initial disturbance, with total size 1000 x 1000 grid nodes and 1000 particles per grid node. V100 cards with 16 Gb of RAM were used for all runs but one (see below). The speed-up is measured 380 as $s_N = t_{\text{ref}}/t_N$. $t$ denotes a computation time, counted from the start of the main algorithm loop to its end, thus excluding the variables' initialisation and output. Each run is completed five times and the average value for each type of run is given

in Figure 7. $t_{\text{ref}}$ is a reference computation time, $t_N$ is the computation time using $N$ processes. The results are given in a log-log representation, to emphasise the behaviour of the code at low and large number of processes. To simplify the analysis and contrary to the usual approach followed for serial processes, $t_{\text{ref}}$ is not chosen as $t_{\text{ref}} = t_1$, but here as $t_{\text{ref}} = t_4$ for the following reason. When using between 4 and 16 GPUs, we achieve the ideal scaling: when doubling the amount of processes, we halve the computation time. In other words all $\{s_4, ..., s_{16}\}$ lie along the straight line of slope 1/4 (since $t_{\text{ref}}$ is taken for four processes). Points for lower and higher number of processes diverge from this ideal scaling. For low numbers of processes, the reason for the relative slow-down is the memory usage of the cards. When dividing the problem between two cards, 87% of their memory is used[1], compared to the 47% usage in the case of four processes. This reflects the fact that GPUs only possess a finite number of parallel threads, though this number surpasses 5000 for this precise hardware. In turn, when an operation needs more threads than available, the computation time is increased. On the opposite end of the test, for $N > 16$, it is the irreducible operations, such as MPI communications or the kernels calls by the CPU, which become greater than the actual calculation time by the GPUs, and lead the speed-ups to diverge from the ideal linear evolution.

The weak scaling of the code is measured by increasing the size of the problem while increasing the number of processes, keeping the same load on single processes. The results are given using the same definition of the speed up, still using the run $N = 4$ for reference. Each process now simulates a 1000 x 125 nodes domain with 1000 particles per node (thus corresponding to the previous $N = 8$ run). Because of the much smaller differences in computation time, the scales are kept linear. As previously, $s_4$ is defined as 1. The GPU memory usage is now 27% and the computation time is 260 seconds for the reference $N = 4$. Using only 1 process results in a speed-up of 1.04, i.e. the runs completes about 4% faster than the reference run. For $N = 2, 3, 4$, the computation time is equal within a second, resulting in a plateau of values around $s_N = 1$. For $N > 4$, another plateau is reached with a speed-up of 0.96, now computing 4% slower than the reference run, independently of how any GPUs are used. The interpretation of these three different values (1.04, 1.00 and 0.96) is straightforward. When running only one process, no MPI communication between processes is necessary, resulting in a faster run. Increasing the number of processes from two to four only involves the exact same amount of communications between neighbouring processes, not affecting the other communications, and the computation time is unchanged. From four to five (and more) processes, communications are now done between two or more computation nodes (with one compute node hosting four GPUs in Jean-Zay cluster used here). Communicating data between cards within a same node is faster than between cards on different nodes, and therefore an observable slow-down happens at $N = 5$. Increasing the number of nodes (from eight cards to sixteen, then to twenty) does not affect the computation time, for the same reason that increasing the number of cards within one node result in the same computation time.

These two tests exhibit a fine behaviour of *Menura*'s MPI parallelisation, also showing that for this problem size, MPI communications do not cost more than 4 to 8 % additional computation time, depending on the number of nodes used.

---

[1]One larger 32 Gb card was used for the case of $N = 1$

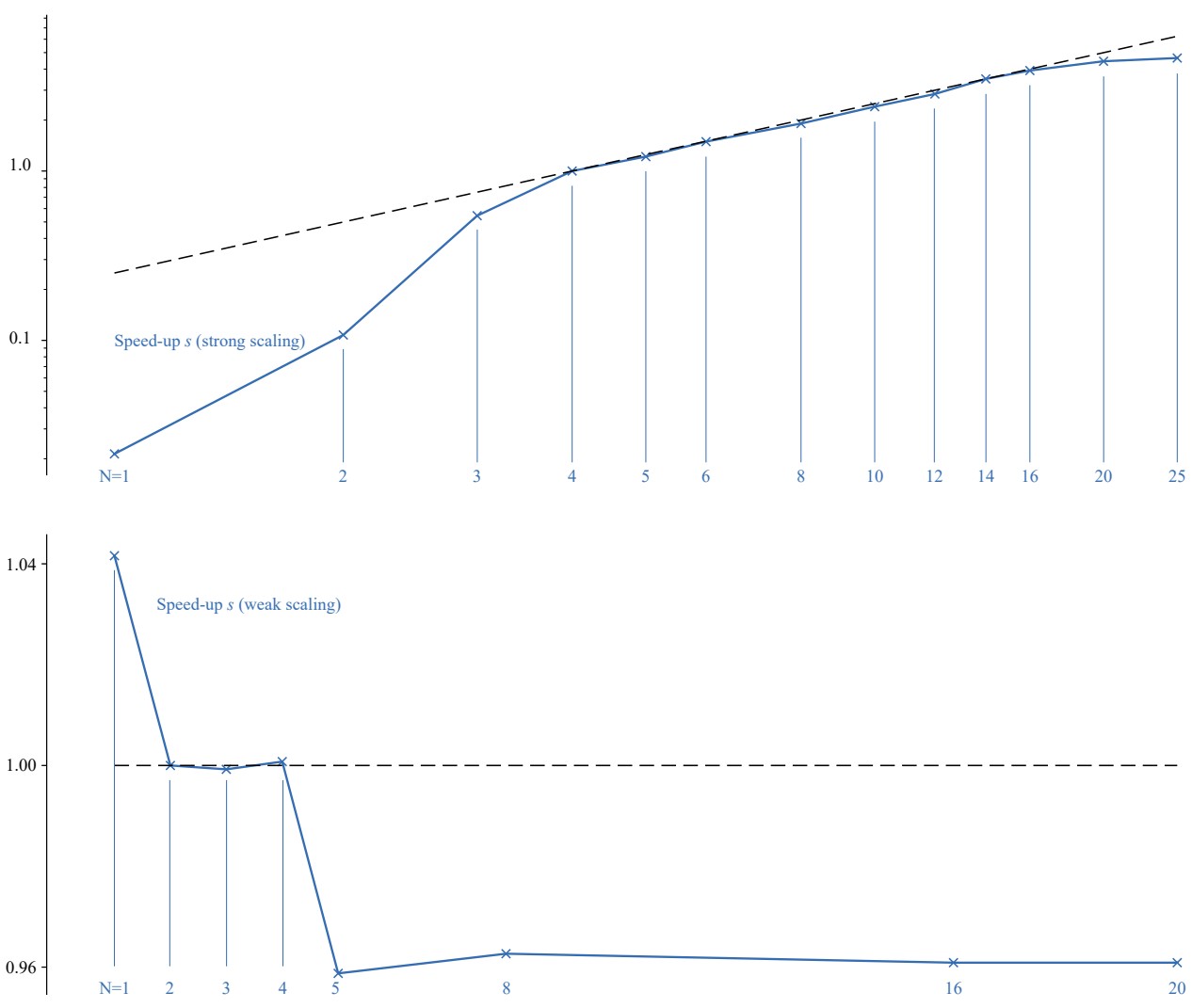

**Figure 7.** Results for the strong (upper panel) and weak (lower panel) scaling properties of *Menura*. The strong scaling test consists of a simulation domain of a 1000 x 1000 grid with 1000 particles per grid node, divided into an increasing number of processes (i.e. GPUs). During the weak scaling test, the load on each GPU is kept constant (1000 x 125 grid, 1000 particles per node) and the number of processes is increased. The speed-up in both cases is defined as $s_N = t_{\mathrm{ref}}/t_N$ with here $t_{\mathrm{ref}} = t_4$.

| | |
|---|---|
| $v_0$ | 363 km/s (6.66 $v_A$) |
| $Q$ | $5 \cdot 10^{26}$ s$^{-1}$ |
| $\nu_i$ | $2 \cdot 10^{-7}$ s$^{-1}$ |
| $u_0$ | 1 km/s |

**Table 3.** Physical parameters of the model comet.

## 6 First result

We now focus on the result of Step 2, in which cometary ions were steadily added to the turbulent plasma of Step 1, moving
at a super-Alfvénic and super-sonic speed. Table 3 lists the physical parameters used for Step 2. After 4000 iterations, the total
number of cometary macro-particles in the simulation domain reaches a constant average value: the comet is fully developed
and has reached an average "steady" state. From this point, we simulate several full injection periods (1500 iterations), looping
over the domain of the injection tank in Figure 6. As an example, Figure 8 displays the state of the system at iteration 6000,
focusing here again on the perpendicular fluctuations of the magnetic field. This time the colour scale is logarithmic, since
magnetic field fluctuations are spanning over a much wider range than previously. While being advected through a dense
cometary atmosphere, the solar wind magnetic field *piles up* (augmentation of its amplitude because of the slowing down of
the total plasma bulk velocity) and *drapes* (deformation of its field lines due to the differential pile-up around the density profile
of the coma), as first theorised by (Alfven, 1957). This general result was always applied to the global, average magnetic field,
and was observed in situ at the various comets visited by space probes.

Without diving very deep in the first results of *Menura*, we see that the pile-up and the draping of upstream perpendicular
magnetic field fluctuations also has an important impact on the tail of the comet, with the creation of large amplitude magnetic
field vortices of medium and small size. This phenomenon, together with a deeper exploration of the impact of solar wind
turbulence on the physics of a comet, are gathered in a subsequent publication.

## 7 Conclusions

This publication introduces a new hybrid PIC plasma solver, *Menura*, that allows for the first time to investigate the impact of
a turbulent plasma flow on an obstacle. For this purpose, a new 2-step simulation approach has been developed which consist
of, first, developing a turbulent plasma, and second, injecting it periodically in a box containing an obstacle. The model has
been validated with respect to well-known fluid and kinetic plasma phenomena. *Menura* has also proven to provide the results
expected at the output of this first step of the model – namely decaying magnetised plasma turbulence.

Until now, all planetary science oriented simulations have ignored all-together the turbulent nature of the solar wind plasma
flow, in terms of structures and in terms of energy. *Menura* has been design to fulfill this deficiency and it will now allow
us to explore, for the first time, some fundamental questions that have remained open regarding the impact of the solar wind
on different solar system objects, such as: what happens to turbulent magnetic field structures when it impacts on an obstacle

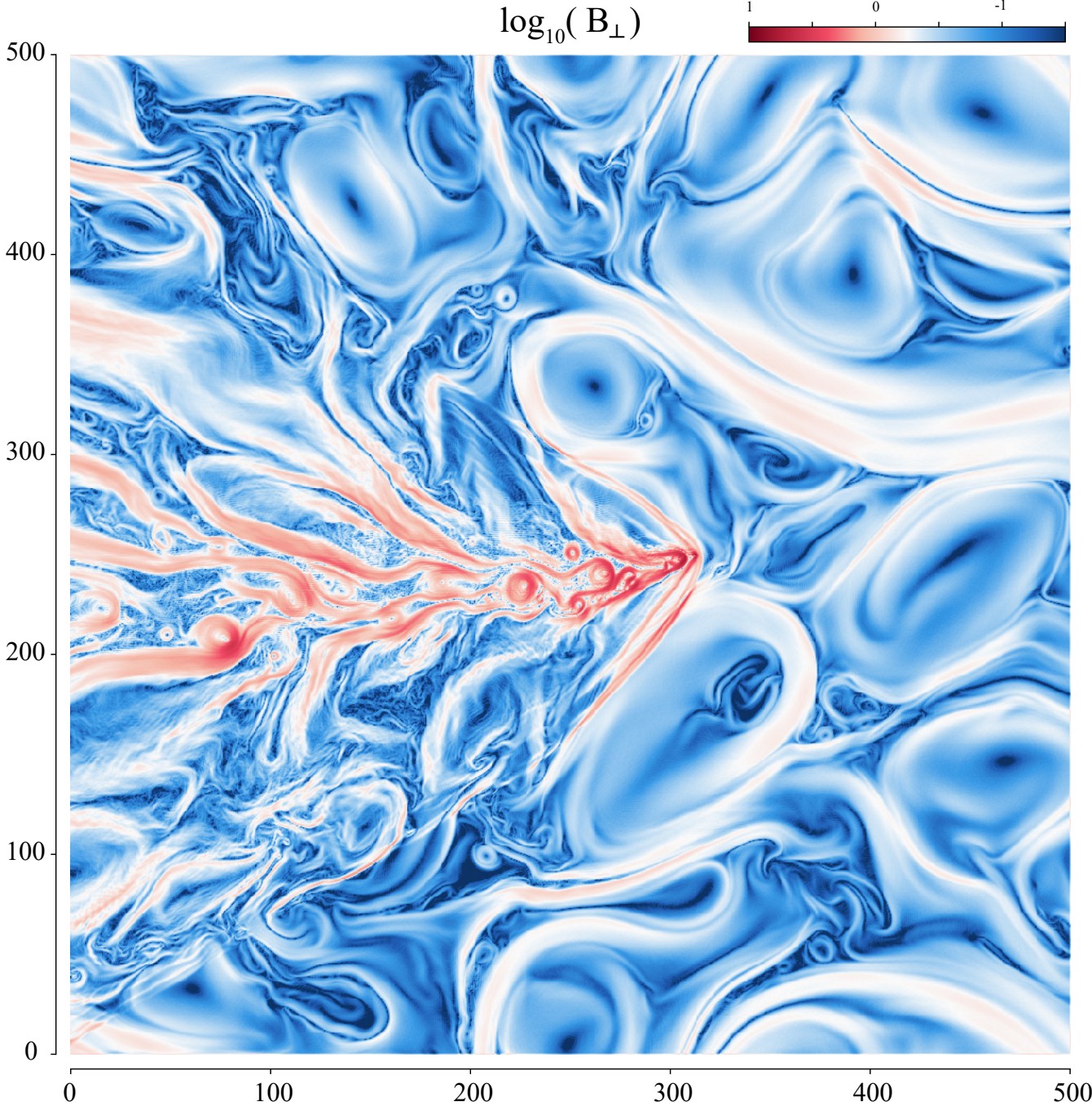

**Figure 8.** Perpendicular magnetic fluctuations during the interaction.

such as a magnetosphere? How are the properties of a turbulent plasma flow reset as is crosses a shock, such as the solar wind crossing a planetary bow shock? How does the additional energy stored in the perpendicular magnetic and velocity field components impact the large-scale structures and dynamics of planetary magnetospheres?

On top of the study of the interaction between the turbulent solar wind and solar system obstacles, we are confident that the new modeling framework developed in this work, in particular its 2-step approach might as well be suitable for the study of energetic solar wind phenomena, namely Co-rotating Interaction Regions and Coronal Mass Ejections, which could be similarly simulated first in the absence of an obstacle, to then be used as inputs of a second step including obstacles.

*Code availability.* The code is open-source and available at https://gitlab.com/etienne.behar/menura, its documentation available at https://menura.readthedocs.io/en/latest/

## Appendix A: Normalised equations

In *Menura*'s solver, all variables are normalised using the background magnetic field amplitude $B_0$ and background density $n_0$, or equivalently using the corresponding proton gyrofrequency $\omega_{ci0}$ and Alfvén speed $v_{A0}$. The background variables definitions were previously given in Table 1. Based on these definitions, one can derive the following main equations of the solver. A normalised variable $\tilde{a}$ is obtained by dividing this variable by its background value, $\tilde{a} = a/a_0$, equivalently $a = \tilde{a}\,a_0$. We first consider the Faraday's law, and using the background parameters definitions of Table **??**:

$$\frac{\partial \mathbf{B}}{\partial t} = -\nabla \times \mathbf{E} \tag{A1}$$

$$\Rightarrow \frac{\partial \tilde{\mathbf{B}} B_0}{\partial \tilde{t}\, t_0} = -\tilde{\nabla}/x_0 \times (\tilde{\mathbf{E}}\, E_0) \tag{A2}$$

$$\Rightarrow \frac{\partial \tilde{\mathbf{B}} B_0}{\partial \tilde{t}/\omega_{ci0}} = -\tilde{\nabla}/d_{i0} \times (\tilde{\mathbf{E}}\, v_{A0}\, B_0) \tag{A3}$$

$$\Rightarrow \frac{\partial \tilde{\mathbf{B}}}{\partial \tilde{t}} = -\tilde{\nabla} \times \tilde{\mathbf{E}} \tag{A4}$$

with

$$\tilde{\nabla} = \left(\frac{\partial}{\partial \tilde{x}}, \frac{\partial}{\partial \tilde{y}}\right) \tag{A5}$$

In other words, the Faraday's law expressed with normalised variables is unchanged compared to its SI definition. The Ohm's law becomes:

$$\tilde{\mathbf{E}} = -\tilde{\mathbf{u_i}} \times \tilde{\mathbf{B}} + \tilde{\mathbf{J}} \times \tilde{\mathbf{B}} + \tilde{\nabla} \cdot \tilde{p_e} - \tilde{\eta_h} \tilde{\nabla}^2 \tilde{\mathbf{J}} \tag{A6}$$

with

$$\tilde{\mathbf{J}} = \tilde{\nabla} \times \tilde{\mathbf{B}} \tag{A7}$$

and

$$\tilde{p}_e = \beta_e \, \tilde{n}^\kappa \tag{A8}$$

Concerning the gathering of particles moments,

$$\tilde{n} = \sum_{\text{spec}} w_{\text{spec}} \sum_p W(\tilde{\mathbf{r_p}}) \tag{A9}$$

with $w_{\text{spec}} = \tilde{n}_{\text{spec}}/\text{particle-per-node}_{\text{spec}}$ . For the solar wind proton, $\tilde{n} = 1$ and one simply gets $w_{\text{sw}} = 1/\text{particle-per-node}$ . $W(\tilde{\mathbf{r_p}})$ stands for the shape factor, triangular in our case (in 2 spatial dimensions, one macro-particle affects the density and current of nine grid nodes, with linear weights).

$$\tilde{J}_i = \sum_{\text{spec}} w_{\text{spec}} \sum_p \tilde{\mathbf{u_i}}(\tilde{\mathbf{r_p}}) W(\tilde{\mathbf{r_p}}) \tag{A10}$$

## Appendix B: $\nabla \cdot \mathbf{B}$ and total energy

Starting with the 2-dimensional Faraday's law (one can ignore the third component, which cannot take part in the divergence of the magnetic field since in 2 dimensions $\partial_z \bullet \equiv 0$),

$$\begin{aligned} \partial_t B_x &= \phantom{-}\partial_y E_z \,, \\ \partial_t B_y &= -\partial_x E_z \,, \end{aligned} \tag{B1}$$

discretised to

$$\begin{aligned} \nabla_t B_{x,\,i,\,j} &= +1/\Delta t \, \nabla_y E_{z,\,i,\,j} \,, \\ \nabla_t B_{y,\,i,\,j} &= -1/\Delta t \, \nabla_x E_{z,\,i,\,j} \,, \end{aligned} \tag{B2}$$

with the notation $\nabla_t, \nabla_x, \nabla_y$ representing the discrete temporal and spatial derivatives. The five-point-stencil central finite difference discretisation of $\partial_y E_z$ reads

$$\nabla_y E_{z,\,i,\,j} = 1/(12\,\Delta x)\,(E_{z,\,i,\,j-2} - 8E_{z,\,i,\,j-1} + 8E_{z,\,i,\,j+1} - E_{z,\,i,\,j+2}) \,. \tag{B3}$$

The divergence of the magnetic field increment $\Delta B$ is then

$$div(\Delta B_{i,\,j}) = \frac{1}{12^2\,\Delta x\,\Delta t}\left(\nabla_x(\nabla_y E_{z,\,i,\,j}) - \nabla_y(\nabla_y E_{z,\,i,\,j})\right) . \tag{B4}$$

The two consecutive finite differences on the electric field component can be expanded to

$$div(\Delta B_{i,\,j}) = \frac{1}{12^2\,\Delta x\,\Delta t}\left(E_{z,\,i-2,\,j-2} - 8E_{z,\,i-2,\,j-1} + 8E_{z,\,i-2,\,j+1} - E_{z,\,i-2,\,j+2}\right)$$

$$\left(-8E_{z,\,i-1,\,j-2} + 64E_{z,\,i-1,\,j-1} - 64E_{z,\,i-1,\,j+1} + 8E_{z,\,i-1,\,j+2}\right)$$

$$\left(8E_{z,\,i+1,\,j-2} - 64E_{z,\,i+1,\,j-1} + 64E_{z,\,i+1,\,j+1} - 8E_{z,\,i+1,\,j+2}\right)$$

$$\left(-E_{z,\,i+2,\,j-2} + 8E_{z,\,i+2,\,j-1} - 8E_{z,\,i+2,\,j+1} + E_{z,\,i+2,\,j+2}\right)$$

$$\left(-E_{z,\,i-2,\,j-2} + 8E_{z,\,i-1,\,j-2} - 8E_{z,\,i+1,\,j-2} + E_{z,\,i+2,\,j-2}\right)$$

$$\left(8E_{z,\,i-2,\,j-1} - 64E_{z,\,i-1,\,j-1} + 64E_{z,\,i+1,\,j-1} - 8E_{z,\,i+2,\,j-1}\right)$$

$$\left(-8E_{z,\,i-2,\,j+1} + 64E_{z,\,i-1,\,j+1} - 64E_{z,\,i+1,\,j+1} + 8E_{z,\,i+2,\,j+1}\right)$$

$$\left(E_{z,\,i-2,\,j+2} - 8E_{z,\,i-1,\,j+2} + 8E_{z,\,i+1,\,j+2} - E_{z,\,i+2,\,j+2}\right)$$

in which terms cancel each-other two by two, resulting in a divergence-free magnetic field increment, $div(\Delta B_{i,\,j}) = 0$. It follows that only round-off errors will accumulate in the time evolution of $div(B)$. The same argument is classically done for constrained transport schemes, which use staggered grids to ensure the same property, with the additional complexity of secondary variables for the electric and magnetic fields, and fields interpolation/averaging between cell centres and cell edges and corners. (Tóth, 2000) provides great insights on the constrained transport and finite central differences schemes, also showing that both conserve the magnetic field divergence.

The time evolution of the divergence of the magnetic field for the decaying turbulence run is shown in Figure B1. We find that the variance of the divergence grows but remains smaller than $10^{-11}$, while the maximum value of the divergence (using its absolute value), remains lower than some $4.\,10^{-5}$, given an initial value of $2.\,10^{-6}$. This growth is due to accumulating round-off errors, over tens of thousand of magnetic field pushes. It was tested that for the exact same problem, increasing the number of time steps increases this accumulated error, despite a finer time resolution.

The total energy, despite a clear decrease over most of the simulation time, is bounded within $+1\%$ and $-4\%$. An additional run was used, which does not include initial perturbation, i.e. an homogeneous plasma. This run shows a nearly perfect energy conservation, with departures of the order of $10^{-5}$ the total energy at initial time.

*Author contributions.* TEXT

*Competing interests.* TEXT

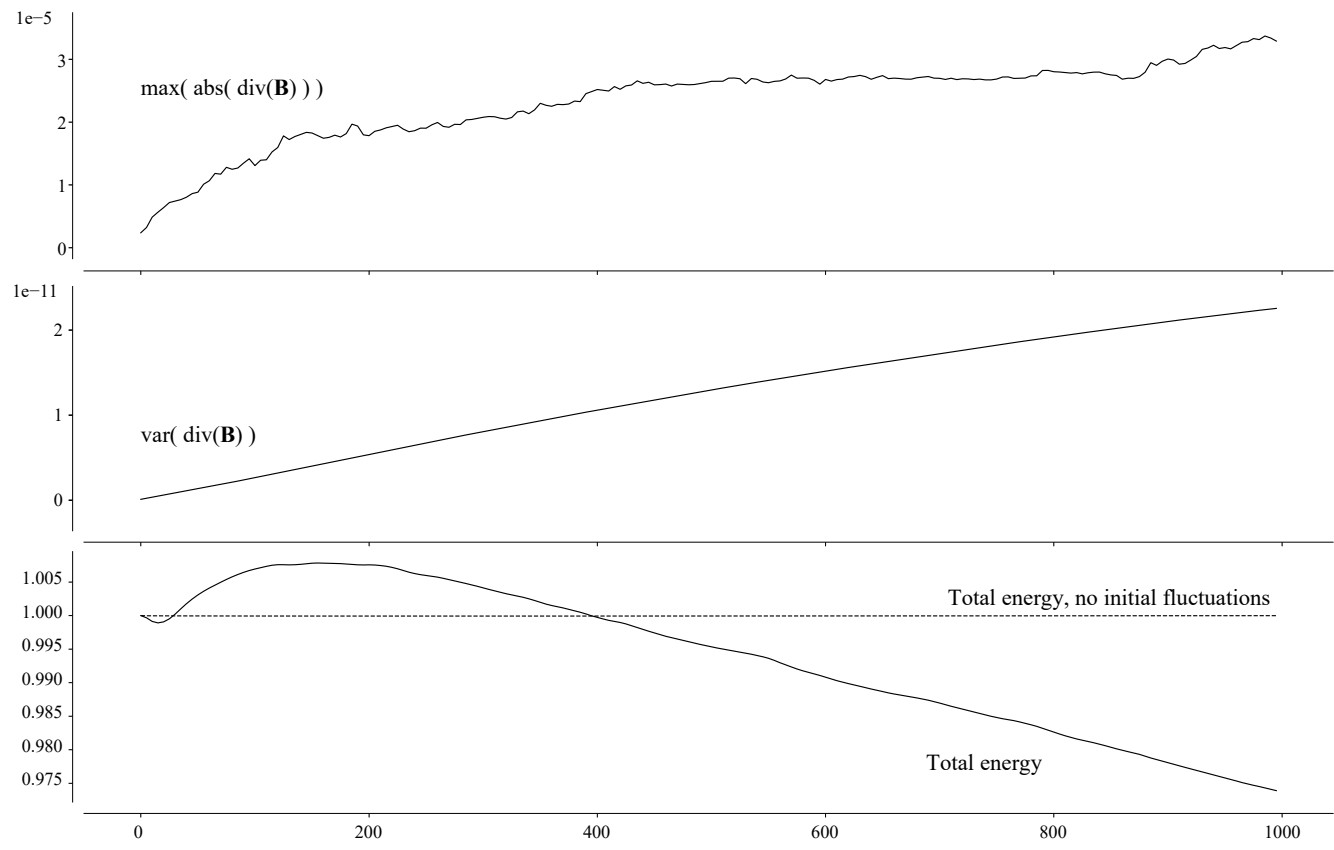

**Figure B1.** Time evolution of the maximum of the divergence of the magnetic field (upper panel) and of the total energy (middle panel) for the decaying turbulence run.

*Disclaimer.* TEXT

*Acknowledgements.* E. Behar thanks Assoc. Prof. Matthew Kunz for the valuable discussion concerning constrained transport in the context of hybrid PIC codes, and acknowledges support from Swedish National Research Council, Grant 2019-06289. This research was conducted using computational resources provided by the Swedish National Infrastructure for Computing (SNIC), Project SNIC 2020/5-290 and SNIC 2021/22-24 at the High Performance Computing Center North (HPC2N), Umeå University, Sweden. This work was granted access to the HPC resources of IDRIS under the allocation 2021-AP010412309 made by GENCI. Work at LPC2E and Lagrange was partly funded by
CNES. S. F. acknowledges supports from the Swedish Research Council (VR) grant 2018-03454 and SNSA grant 115/18.

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
