# Peer review of "Menura*: a code for simulating the interaction between a turbulent solar wind and solar system bodies"

_Annales Geophysicae, 2021_

## Author Response (AR1)

Reply RC1

We thank the referee for the encouraging feedback and for the time spent on the draft. We try to tackle the comments in the following answer.

1 – Readability of the introduction : We have shortened and tried to fix some language issues, though we can iterate once more if needed.

2 – References formatting : that is our mistake, \cite were replaced by \citep everywhere.

3 – Figure reference: fixed.

4 – The polytropic index k is now tackled with the following sentence "In all the results presented below, an index of 1 was used, corresponding to an isothermal process." when introducing equation 6. The change of index when discussing the magnetosonic mode now reads

"The magnetosonic modes were also tested using a different polytropic index of 5/3 instead of 1, resulting in a shift of the dispersion relation along the $\omega$-axis. Changing the polytropic index in both \emph{Menura} and \emph{WHAMP} resulted in the same agreement."

Additionally, at the beginning of the physical test, it is now stated "A polytropic index of 1 is used here, with no resistivity."

5 – The paragraph tackling the hyper-resitivity has been expanded to introduce the corresponding dissipative scale. The value of 0 was mentioned above for the physical tests, and the value and corresponding scale for the turbulent run also given.

", introducing the Laplacian of the total current and the hyper-resistivity coefficient, $\eta_h \nabla^2 \mathbf{J}$. The dissipative scale $L_\text{dis}$ of such a term is characterised by the physical time of the simulation $T = \text{nb. iterations} \times dt$ and the hyper-resistivity, such as $L_\text{dis} = (\eta_h \cdot T)^{1/4}$."

"The polytropic index is 1 and a normalised hyper-resistivity of $\eta_h=1.5 \cdot 10^{-3}$ is used, corresponding to a dissipative scale at time $t=500\ \omega_{ci0}^{-1}$ of 0.93 $d_i$, i.e. the scale of the smallest fluctuations simulated with a node spacing of $\Delta x = 0.5 d_i$."

As stated previously, all parameters including the polytropic index and eta_h are kept unchanged for step 2 of the simulation.

6 – Test done on a 2D domain with one main direction x: it is indeed a cut taken for one precise index y.

"(saving one cut, given by one single index along the $y$-direction)"

7 – The number of particle per node is added for the ion acoustic Landau damping.

"This low amplitude, allowing for comparison with the linear solver, further increase the need for a high number of particle per node, so the 1\% oscillation in number density can be resolved by the finite number of particles. For this run, 32768 ($2^{15}$) particles per grid node were used."

8 - k in units of di0^{-1}, rather than di0: absolutely.

9 – Scalability: a fairly complete study of scalability was motivated by the comment, which is now given in paragraph 5.4.2 and Figure 7.

10 & 11 – Table A1 was removed, and was indeed an old copy of Table 1. The caption now reads

"Background values used to normalise all variables in the solver (cf. Eq. \ref{eq:norm})"

All typos were addressed with the exception of one, we kept "to go further in the in situ space data analysis, further in their understanding …"

Reply RC2

We thank the referee for the attention and time spent on the draft, and for the insightful feedback we received. We try our best to tackle all comments in the following answer.

Top-level

1 & 2 : These two comments are relevant and brought us back to some fundamentals concerning the solver. We have in the meantime implemented a constrained transport version of the solver (using a Yee meshing), but it turns out that for exactly the same argument classicaly derived for the constrained transport, the magnetic field pushes done using central finite difference are also divergence-free. This is litteraly the same derivation as for the constrained transport. The tremendous advantage in our case is that we only work with one single grid, with no need for interpolation/averaging between cell centres, corners and faces. Even more interesting, this divergence-free property applies to our main and single magnetic field, and not to its face-centred version used in constrained transport. This whole topic is also very well covered by Toth 2000, a reference we have added. We now show how our discretisation conserves div(B) in the limit of accumulating round-off errors in Appendix B, which also contains a plot of the time evolution of the variance and maximum of div(B).

3 – We have added the following sentences to better introduce the obstacle we consider.

"A comet nucleus is from a few to a few tens of kilometers large, with a gravitational pull not strong enough to overcome the kinetic energy gained by the molecules during sublimation."

" "
→ "Considering the spatial resolution of the simulation, the solid nucleus can be neglected. By also neglecting the gravitational force on molecules as well as any intrinsic magnetic field, …"

4 – We have added a paragraph concerning the simplified physico-chemistry, another relevant missing point:

"One additional simplification is to limit the physico-chemistry of the cometary environment to photo-ionisation, thus neglecting charge exchanges between the solar wind and the coma, as well as electron impact ionization. Both processes can significantly increase the ionisation of the neutral coma \citep{simonwedlund2019aa}. A global increase or a local change in the production profile is not expected to impact the initial main goal of the model, which is to simulate the turbulent nature of the solar wind during its interaction with an obstacle. We note however that the influence of upstream turbulence on the physico-chemistry of an obstacle is yet another promising prospect for the code."

5 – Varying the flow-aligned component of the magnetic field: this is a very exciting comment about a point we entirely missed. We have added the following paragraph to section 5.2:

"Another major advantage of working in the solar wind reference frame is the possibility to simulate magnetic field variations in all directions, including the relative plasma-object direction. For studying the interaction between Co-rotating Interaction Regions and an object for instance, one need to vary the direction of the magnetic field upstream of the object of interest. In the object reference frame, such a temporal variation of the magnetic field is frozen-in the flow and advected downstream through the simulation domain by the convective electric field. Considering the ideal Ohm's law $\mathbf{E} = -\mathbf{u}_i \times \mathbf{B}$ and Faraday's

law $\partial_t \mathbf{B} = -\nabla \times \mathbf{E}$, and considering a plasma flowing along the $x$-axis $\mathbf{u}_i=u_i\hat{x}$, we get the time evolution of each magnetic field component as

$$
\begin{align*}
\partial_t B_x &= \partial_y(u_i B_y) - \partial_z(u_i B_z) \\
\partial_t B_y &= -\partial_x(u_i B_y) \\
\partial_t B_z &= \partial_z(u_i B_z) \\
\end{align*}
$$

The direct implication of this system is that any temporal variation we may force on the upstream $B_x$ cannot have a self-consistent influence on the time evolution of the magnetic field elsewhere: only variations forced along magnetic field components perpendicular to the flow direction can be advected downstream, through this ideal frozen-in condition. In contrast, when working in the solar wind reference frame, we can impose spatial fluctuations of the magnetic field (equivalently temporal in the object frame) in all directions: in this frame these fluctuations are not being advected, it is rather the object itself moving through the fluctuations. This effectively removes the constraint on flow-aligned variation of the magnetic field, opening promising possibilities for the simulation of various solar wind events, such as CIRs or sector boundary crossings."

Line-by-line comments:

Line 1: "between planetary science and fundamental plasma physics", perhaps, to more specifically refer to turbulence studies as a part of fundamental plasma physics? I would argue plasma physics and planetary science have been bridging for quite a while with e.g. studies of Mars' ion escape driven by solar wind.

We would like to keep "fundamental plasma physics/phenomena" (mostly microphysics of wave-particle interacgtion, instabilities, turbulence, etc) seperated from planetary/mangetospheric physics, including atmospheric erosion. We agree that both fields have been overlapping for some time. This part of the introduction was re-worked based on Referee #1 comments, we hope the current version is satisfactory.

Line 63: "running its solver exclusively on GPUs" – to state this already at this point: on multiple GPUs in parallel

Yes, added.

Lines 72-75: Esp. considering the large macroparticle counts required for the turbulence simulation, the authors should note the hybrid-Vlasov methods and briefly discuss the choice of PIC over Vlasov.

Vlasov and Hybrid-Vlasov is the exciting future, for sure! However there are two main limitations that make these models still out of our reach. The first one is dealing with boundary conditions, and all planetary specificities (like adding planetary ions). The solutions might exist for all our purposes but are not yet extremely widespread, so we would have to "innovate" on both fronts. Or in other words, PIC codes seems still much more flexible than Vlasov codes. The other is worse, especially for GPU based codes; there is still two orders of magnitude in terms of memory needed, from hybrid PIC to hybrid Vlasov. Our 1000x1000 domain with 2000 ppc results in 50Gb memory, devided between 16 or 32 Gb devices. The same domain with say 50x50x50 VDF resolution (used in the litterature) adds up to 3 Tb. To this extent we would not agree that we had to choose between the two methods, unfortunately.

Line 75: "evaluated at the nodes", esp. footnote: Depending on the context, values evaluated at "cell" locations may refer to values averaged over the volume of the cell, instead of point-wise values at the nodes (which is less ambiguous, as nodes usually refer to geometric points, in this context), so "equivalently used by other authors" is not true, in general. This is also not relevant for the manuscript, so I would drop the footnote.

Absolutely. We have removed the foot-note.

Figure 1: E* does not appear to be used for J_n+1 as stated in the text.

Indeed, we have added the missing arrow.

Lines 185-188: wavenumber and inertial length dimensions are incompatible – is this a formatting error with a missing slash? I am confused by the statement "inertial range" – is this referring to the ion inertial length (in which case I'd expect k ~ di) or Inertial Alfvén Waves?

All our k values indeed had the wrong dimension in the text, now fixed. Concerning the inertial range, it is the denomination used in turbulence studies of the -5/3-slope magnetic fluctuations spectrum, (typically from 1e-3 to 1 Hz in the solar wind, bounded between the

correlation length and the ion length d_i or rho_i), where the energy cascade is dominated by fluid MHD physics (as for instance illustrated and discussed in the classical publication of https://doi.org/10.1098/rsta.2014.0155 ). We have added this reference.

Line 179: Adding to particle velocities before the particle distributions are introduced as Maxwellian is slightly out of order.

We have re-worked the two sentences, now reading

"are added to both the homogeneous background magnetic field $\mathbf{B}_0$ and the ion bulk velocity $\mathbf{u}_i$. Particle velocities are initialised according to a Maxwellian distribution, with a thermal speed equal to one Alfv\'en speed, and a bulk velocity given by the initial fluctuation"

Lines 200-205: Do the authors mean to introduce a Fourier transform in (x,y) instead of (\perpendicular,\parallel) plane? It would not make much sense to do a Fourier transform in the B\parallel direction, if the guide field is out-of-plane. Then "chosen bins of k_\perp" would make sense in the (k_x, k_y) plane. The k_\perp binning is said to be arbitrarily chosen – is this an logarithmic binning with an arbitrary number of bins, or a truly arbitrary function in k\perp? This should be specified more exactly.

Absolutely, (x, y)-plane instead of (para, perp)-plane. That was an embarassing typo left from some older 3D considerations.

The width of the rings is constant through all scales, resulting in the coarser spectra at low frequencies, because of the log visualisation. We have added "The width of the rings, constant through all scales,"

Line 216: initial bulk velocity perturbation spectrum is not shown after all?

The sentence might have been misleading: we use at low frequencies the initial magnetic field fluctuations, and at high frequencies the initial velocity fluctuations to evaluagte the PIC noise.

"Additionally, the initial spectra of the magnetic field and bulk velocity perturbations are over-plotted, to show where the energy is injected in the lower spatial frequencies (using the magnetic field fluctuations), and the level of noise introduced by the finite number of particles per node used, at high frequencies (using the bulk velocity field)."

Line 231: "The first and foremost interest… is it size" is not well-qualified: The size of the plasma environment might be of foremost interest to plasma scientists, while e.g. the size of the nucleus for others (and this is hardly distinguishable from water ion gyroradii).

"its size" changed to "size of its plasma environment"

General Editing:

Citations are not well formatted in the text (\cite vs \citep?), making the text hard to parse.

> \citep is now used throughout the text.

Line-by-line editing

Line 3: overlapping field*s*

> Done

Lines 14-15: a bit repetitive with 3x "communities"

> Only one now in the new phrasing.

Line 16: eventful instead of event-full

> Yes.

Lin 27: "planet*ary* magnetospheres"

> Ok.

Line 36: "shedding new light"

> Yes indeed.

Line 46: "magnetosphere"

> Yes.

Line 101: "copied to"?

> Ok.

Line 105: "field values"

> Ok.

Line 115: "one gets"

> Absolutely.

Line 116: "three types of variables", perhaps, since there are a lot of particles, even at single cell-level.

> Ok.

Line 142 "normalized equations of the solver are given in *Appendix* A"

> Yes.

Table 1 Caption missing.

> Added.

Line 150: Introduce panel labels for Fig.2, with the B-parallel Fourier transform as panel a) and perpendicular as panel b) and use these in the text.

> "Left-hand panel: Alfv\'enic modes, $B_0$ taken along the main spatial dimension. Right-hand panel: Magnetosonic modes, $B_0$ taken perpendicular to the main spatial dimension."

"(Alfv\'enic fluctuations to the left, magnetosonic to the right)"

Line 153: format k.d_i with \cdot or other more suitable operator

Ok

Figure 2: Missing subscripts for k (parallel, perpendicular); missing panel labels

Done

Line 162: "not captured"

Yes.

Line 164: "both growth rates", perhaps

Yes.

Figure 3, left: should the WHAMP solution label be |E| for compatibility?

Graphical typo, yes.

Line 171: "the amount of particles per node"

The precise number was added in the mean time.

Line 181: remove "while"

All right.

Line 186: k_inj,min/max formatting missing commas? For wave vector amplitude, maybe use wave vector length instead, to contrast with perturbation amplitude.

Ok, "values" used instead of second "amplitude".

Figure 4 requires legend and colormap for left panel, right panel could also include slopes for high-k analytic lines

Sure.

Line 213: "associated with"

Yes.

Figure 5: missing units

Indeed, these are normalised field values.

Line 256: "Small scale oscillations are current"?

Common

Line 262: "macroparticles"

Ok

Line 295: "Message Passing Interface"

Yes

Line 307: Perhaps "injector data structure", rather than "variable"

Sure.

Line 309: "NVIDIA"

Yes.

Table 3: u0 is given twice, "Physical parameters of the model comet", since an actual comet is more than the model includes

$u_0$ for solar wind speed should have been $v_0$.

Ok sure.

Line 331: "one full turbulence domain injection period: 1500…" or so? Current form implies 1500 iterations being more than one full period.

" From this point, we simulate several full injection periods (1500 iterations), looping over the domain of the injection tank in Figure"

Line 350: "this lack" - "this deficiency"?

Mucb better yes.

Line 353: "reset" is a strange word here. "Processed", perhaps?

We keep reset here, in the sense of "setting again".

Line 358: "phenomena"

Yes.